# Do Changes in *ACE-2* Expression Affect SARS-CoV-2 Virulence and Related Complications: A Closer Look into Membrane-Bound and Soluble Forms

**DOI:** 10.3390/ijms22136703

**Published:** 2021-06-23

**Authors:** Huseyin C. Yalcin, Vijayakumar Sukumaran, Mahmoud Khatib A. A. Al-Ruweidi, Samar Shurbaji

**Affiliations:** 1Biomedical Research Center, Qatar University, Doha 2713, Qatar; ma1207471@student.qu.edu.qa (M.K.A.A.A.-R.); ss1104227@student.qu.edu.qa (S.S.); 2Department of Chemistry and Earth Sciences, College of Arts and Sciences, Qatar University, Doha 2713, Qatar

**Keywords:** *ACE-2*, SARS-CoV, COVID-19, shedding, ACE inhibitors, RAAS, angiotensin receptor blockers, ARDS

## Abstract

The SARS-CoV-2 virus utilizes angiotensin converting enzyme (*ACE-2*) for cell entry and infection. This enzyme has important functions in the renin-angiotensin aldosterone system to preserve cardiovascular function. In addition to the heart, it is expressed in many tissues including the lung, intestines, brain, and kidney, however, its functions in these organs are mostly unknown. *ACE-2* has membrane-bound and soluble forms. Its expression levels are altered in disease states and by a variety of medications. Currently, it is not clear how altered *ACE-2* levels influence *ACE-2* virulence and relevant complications. In addition, membrane-bound and soluble forms are thought to have different effects. Most work on this topic in the literature is on the SARS-CoV virus that has a high genetic resemblance to SARS-Co-V-2 and also uses *ACE-2* enzyme to enter the cell, but with much lower affinity. More recent studies on SARS-CoV-2 are mainly clinical studies aiming at relating the effect of medications that are thought to influence *ACE-2* levels, with COVID-19 outcomes for patients under these medications. This review paper aims to summarize what is known about the relationship between *ACE-2* levels and SARS-CoV/SARS-CoV-2 virulence under altered *ACE-2* expression states.

## 1. Introduction

The way a virus can infect an organism is via binding to its specific surface receptors on the cell. Hence, the expression and tissue-specific distribution of such receptors determine the pathogenesis of the disease. Similar to the original severe acute respiratory syndrome coronavirus (SARS-CoV), SARS-CoV-2 utilizes the peptidase angiotensin-converting enzyme 2 (*ACE-2*) for cell entry [1]. Therefore, *ACE-2* expression in different tissues and *ACE-2*’s interaction with SARS-CoV-2 are critical for the progression from early infection to severe coronavirus disease 2019 (COVID-19). For the majority of cases, COVID-19 patients present typical symptoms of inflammation in the lung, and acute respiratory distress syndrome (ARDS) in severe cases, showing that the lung is the primary target of the virus. Lung alveolar type 2 (AT2) cells highly express *ACE-2*, which explains the susceptibility of the organ to SARS-CoV-2 infection. However, the function of this prominent *ACE-2* expression in the lung other than enabling virus entry is not clear. As an important member of the renin-angiotensin aldosterone system (RAAS) pathway, *ACE-2* is a transmembrane protein best known for its homeostatic role in counterbalancing the effects of ACE and Angiotensin II (ANG-II) on the cardiovascular system. *ACE-2* can be found as membrane-bound or soluble forms, both of which can actively cleave ANG-II. Considering *ACE-2*’s primary function is ANG-II cleavage, the reason for higher expression in pulmonary epithelial cells compared to in pulmonary endothelial cells in unclear. Furthermore, *ACE-2* is also highly expressed in cardiomyocytes, kidneys, intestines, male reproduction organs, and the brain [2]. It is unclear whether the non-pulmonary expression of *ACE-2* and the relevant viral infection is responsible for other clinical complications of SARS-CoV-2, and this requires further investigation. Some of the symptoms, such as loss of smell and gastrointestinal complications, suggest SARS-CoV-2 infection in *ACE-2* expressing tissues in COVID-19. Tissue-specific *ACE-2* expressions, as well as circulating soluble concentrations, were shown to be altered in diseased states such as viral infection and cardiac disease [3,4]. Furthermore, certain anti-hypertensive medications are known to alter *ACE-2* levels [5]. Therefore, there is a great interest in revealing a mechanistic relationship (if it exists) between *ACE-2* levels and SARS-CoV-2 virulence/related complications, and how this relationship is affected in diseased states. This review aims to summarize what is known about this relationship and the critical points to be investigated in the future.

## 2. SARS-CoV-2 and *ACE-2*

Coronavirus entry into cells via membrane-bound *ACE-2***:** COVID-19 has become a deadly pandemic affecting all countries. A novel coronavirus, previously named 2019-nCoV, and now known as SARS-CoV-2, was discovered to be the cause of the disease [6]. A coronavirus has spike proteins, commonly called S-protein, distributed on its lipid layer. The virus binds to the specific surface receptors on the target cell’s membrane for cellular entry, leading to replication and infection [3]. Coronavirus S-protein has two functional units, S1 and S2. While S1 contains the receptor-binding domain (RBD) for binding to *ACE-2,* which is the SARS-CoV-2’s host receptor, the function of S2 is fusion into the host cell membrane. Following the binding of the virus to *ACE-2* by S1, the cleavage site of S2 is exposed and cleaved by the host protease, which was shown to be a critical step for virus infection [3]. While the S-protein homology between SARS-CoV-2 and SARS-CoV was relatively low, with a similarity of amino acid sequence 76.47%, S-protein RBD domains were highly similar, resulting in close van der Waals bond and electrostatic properties in the interaction interface, which shows the acting receptor of SARS-CoV-2 was the same as that of SARS-CoV. Both the original SARS-CoV and recent SARS-CoV-2 utilize *ACE-2* as their host receptor. However, the affinity of SARS-CoV-2 S-protein against *ACE-2* is about 10–20 times higher than that of SARS-CoV S-protein, which is likely to explain highly infectious characteristics of SARS-CoV-2. This high affinity raised the attention to *ACE-2* enzyme, its tissue specific expressions and up-/downregulation in its different forms.

The human *ACE-2* gene contains 18 exons located on the short arm of the X chromosome. Its molecular weight is 120 kD encoding 805 amino acids [7]. As a transmembrane protein, *ACE-2* consists of a signal peptide at the amino-terminal, a single metalloproteinase active site, a transmembrane domain, and a small cytoplasmic domain at the carboxyl terminus. Similar to ACE, *ACE-2* is a cleavable enzyme that releases its extracellular portion into the circulation as soluble *ACE-2*, a mechanism known as shedding. Transmembrane and cytoplasmic domains are missing in the soluble *ACE-2* but it still has an enzymatic activity [6].

*ACE-2* mediated coronavirus entry into the host cell can happen in two independent ways (Figure 1). In the first way, the virus enters the cell via endocytosis. Here, when the virus docks onto *ACE-2*, the catalytic extracellular domain of *ACE-2* is cleaved off, controlled by a specific protease, ADAM17, resulting in internalization of the transmembrane domain. This is followed by the fusing of the viral particles with the assistance of Clathrin and the transport of the virus from the cell membrane to the cytoplasm, aided by the intracellular structure of *ACE-2* [8]. The second way of cell entry is membrane fusion, again mediated by *ACE-2* and also by transmembrane serine protease 2 (TMPRSS2). Here, following the binding of the SARS S-protein to *ACE-2*, processing by TMPRSS2 allows direct fusion at the cell surface [9]. The critical role of TMPRSS2s for SARS-CoV-2 infection was recently confirmed [10]. While TMPRSS2 and ADAM17 compete for processing *ACE-2*, only cleavage by TMPRSS2 enhances SARS-S protein-driven entry [6].

### 2.1. Functions and Distribution of ACE-2 in the Body

Tissue-Specific *ACE-2* expression: SARS-CoV-2 is accepted as an airborne transmitted respiratory virus. Therefore, the virus must initially infect the respiratory tract cells. In situ RNA analysis and single-cell sequencing of the human respiratory tract revealed that *ACE-2* is highly expressed in nasal epithelial cells and lesser expression in bronchial epithelial cells and AT2 cells [11]. In vitro studies were consistent with that finding: upper respiratory tract cells were more permissive to SARS-CoV-2 infection than lower respiratory tract cells [1]. Recently, Zhao and colleagues revealed that, while *ACE-2* is expressed only in 0.64% of pulmonary cells, the majority of the expression is concentrated on AT2 cells (about 83%). Therefore, SARS-CoV-2 seems to efficiently utilize AT2 cells for reproduction and spread. The high AT2 expression of *ACE-2* may be the reason for severe alveolar injury following the initial infection. Previous studies demonstrated that, as well as the lung, *ACE-2* is widely distributed in multiple other tissues including the heart, vascular smooth muscle cells, kidney, gastrointestinal tract, testis, adipose tissue, and brain tissue [3]. *ACE-2* is expressed mainly in the vascular endothelial cells of these organs [12]. Xu and colleagues performed RNA-Seq for 14 organs and showed that *ACE-2* is highly expressed in (from highest to lowest) the colon, gallbladder, heart muscle, kidney, epididymis, breast, ovary, lung, prostate, esophagus, tongue, liver, pancreas, and cerebellum [13] (Figure 2). The same study revealed high *ACE-2* expression in oral cavity epithelial cells, consistent with initiation of infection at the upper respiratory tract. Hikmet and colleagues analyzed the protein expression profiles in a variety of human tissues. *ACE-2* was mainly expressed in renal tubules, intestinal enterocytes, cardiomyocytes, vasculature gallbladder, male reproductive cells, ductal cells, placental trophoblasts, and eye [14]. Interestingly, lung *ACE-2* expression was low with expression constricted to mainly AT2, cells consistent with other studies. Cellular distribution of *ACE-2* in these tissues was constricted to mainly in endothelial cells, epithelial cells, and immune cells such as macrophages [15]. Expression profiles partly reflect the SARS-CoV-2 infection profile, such as common gastrointestinal tract infections with detectable viral RNA in the stool for about 30% of COVID-19 patients [1]. Acute kidney injury and heart injury are common in COVID-19, consistent with high *ACE-2* expression in these organs [6,16].

### 2.2. ACE-2 in the RAAS Pathway

The function of *ACE-2* in organs other than the cardiovascular system is unclear. In the cardiovascular system, the *ACE-2* enzyme performs an important function in the RAS pathway. *ACE-2* is a close homologue of human ACE. A homeostatic counterbalance between ACE and *ACE-2* is important to maintain normal organ physiology, and also can influence the pathogenesis of various diseases [17]. ACE cleaves and converts ANG-I to ANG-II. When ANG-II binds to ANG-II receptor 1, it induces a variety of local and systemic effects on the cardiovascular system, including vasoconstriction, fibrosis, and salt retention (Figure 3). *ACE-2* on the other hand exerts opposite effects in RAAS. By catalyzing the conversion of ANG-I into ANG-(1–9) and ANG-II into ANG 1–7, *ACE-2* induces vasorelaxation, anti-oxidative action, anti-inflammation, and cardioprotection [12]. Therefore, while ACE expression increases ANG-II production, *ACE-2* causes ANG-II downregulation. Decreasing ANG-II via *ACE-2*-ANG 1–7 axis is an important therapeutic approach for cardiovascular disorders (Figure 3). For example, angiotensin receptor blockers (ARBs) and angiotensin-converting enzyme inhibitors (ACEi) are used in cardiac disorders including hypertension, to control ANG-II levels [18].

### 2.3. ACE-2 in Disease

#### Membrane-Bound and Soluble forms of ACE-2, and Mechanism of Shedding

As a transmembrane glycoprotein enzyme, *ACE-2* is composed of a short cytoplasmic domain, a transmembrane domain, a catalytic ectodomain, and an amino-terminal signal peptide. Similar to most other transmembrane proteins, *ACE-2* undergoes proteolytic cleavage from the cell surface to release its catalytic ectodomain part as a soluble form, a process known as ectodomain shedding. Shedding is a key post-translational modifications mechanism for transmembrane proteins such as *ACE-2*. Therefore, *ACE-2* can be found as membrane-bound or as soluble/circulating within the body [19]. Because the catalytic site of *ACE-2* is located in the ectodomain site, the release of the catalytically active part of the enzyme following shedding can influence the enzymatic activities in the cellular microenvironment, but also can influence distal tissue once the soluble enzyme migrates to distal organs via systemic circulation. Additionally, since it is a surface receptor, intracellular signaling and gene expression can be initiated once a ligand binds to it [20] or as a response to a viral infection such as SARS-CoV [21]. Therefore, the ectodomain of *ACE-2* plays an important role in its function, and ectodomain shedding could influence both *ACE-2*-expressing cells/tissues and other neighboring cells/tissues. Shedding of the ectodomain part of *ACE-2* and the release of the soluble form were shown to exist in various diseases including heart disease, kidney disease, lung infection, and ARDS [19].

### 2.4. ACE-2 Shedding and Viral Infection

To improve therapies, it is important to understand how *ACE-2* shedding, as well as membrane-bound and soluble *ACE-2* levels, affect SARS-CoV-2 infection. Li and colleagues were the first groups showing that the S1 domain of the SARS-CoV S-protein binds efficiently to *ACE-2* isolated from SARS-CoV-permissive Vero E6 cells [21]. When a major portion of the cytoplasmic domain of *ACE-2* was deleted, there was no effect on S-driven infection, suggesting that the cytoplasmic domain is not critical for receptor function for infection. Another important finding was that a soluble form of *ACE-2* could successfully block the association of the S1 domain with the cells. Viral replication was inhibited when the cells were cultured in the presence of an anti-*ACE-2* antibody. Hofmann and colleagues showed that *ACE-2* expression levels in various cell lines directly correlated with the susceptibility to SARS-CoV S-driven infection. They also demonstrated that pre-incubation of cell cultures with soluble *ACE-2* ectodomain inhibits SARS-CoV S-driven infection in a dose-dependent manner, further confirming that soluble *ACE-2* can effectively block the binding of SARS-CoV S-protein to membrane-bound *ACE-2* [22]. These findings directed interest to mechanisms for *ACE-2* shedding, which would increase soluble *ACE-2* levels and inhibit viral infection. Lambert and colleagues investigated the involvement of several ADAM family proteins, which largely mediate the shedding of membrane proteins [23]. Ablation of ADAM17 cells in HEK293 and Huh7 cells resulted in reduced *ACE-2* shedding, whereas ADAM17 overexpression significantly increased shedding, providing direct evidence for ADAM17’s direct involvement in regulating the ectodomain shedding of *ACE-2*. Haga and colleagues showed that shedding of the ectodomain of *ACE-2* depends on the TNF-alpha-converting enzyme (TACE) and in contrast to a study by Li and colleagues, they found that cytoplasmic tail is required for shedding and the process was accompanied by TNF-alpha production and tissue damage [24]. In relevance to SARS-CoV, knockdown of TACE or deletion of the cytoplasmic domain of *ACE-2* blocked infection of the virus. These findings provide evidence for modulation of TACE activity by SARS-CoV through *ACE-2*’s cytoplasmic domain. In addition to facilitating the viral entry, the same domain causes tissue damage via the production of TNF-alpha.

Using recombinant SARS-S protein, Glowacka and colleagues showed that SARS-S efficiently engages to *ACE-2* on Vero-6 cells, correlating with efficient induction of *ACE-2* shedding. Binding of the viral S-protein resulted in the shedding of soluble *ACE-2* into the cellular supernatants, and also reduced membrane-bound *ACE-2* expression, which suggests that the shedding of *ACE-2* contributes to the downregulation of *ACE-2* expression in the context of SARS-CoV infection [25]. A following study showed that *ACE-2* can also be processed by TMPRSS2, and it was suggested that *ACE-2* cleavage via TMPRSS2 might increase SARS-S-mediated cellular entry [26]. Type II transmembrane serine proteases (TTSPs) are mainly found on the surface of airway epithelial cells [27]. These were shown to facilitate the entry of multiple respiratory viruses, including human influenza viruses [28] and SARS-CoV [29]. The authors evaluated several TTSPs in their ability to activate S protein entry into 293 T cells and identified that TMPRSS2 enhanced S-mediated entry, more than other tested TTSPs. Since TMPRSS2 is highly expressed in epithelial cells lining the nose, trachea, and distal airways, including type I and type II alveolar cells, TMPRSS2 might play role in lower-airway SARS-CoV and SARS-CoV-2 infections.

In a more recent study, Heurich and colleagues studied and compared TMPRSS2 and ADAM17 in *ACE-2* cleavage [9]. They found that SARS-CoV infection is facilitated by TMPRSS2 via two independent mechanisms: while *ACE-2* cleavage promotes viral uptake, cleavage of SARS-S activates the protein for membrane fusion. They also demonstrate that ADAM17 competes with TMPRSS2 for *ACE-2* processing, but SARS-S-driven entry was not modulated by ADAM17 activity. In addition, the study demonstrated that TMPRSS2 does not facilitate *ACE-2* shedding and even interferes with ADAM17-regulated *ACE-2* shedding, suggesting that ADAM17 and TMPRSS2 cleave *ACE-2* at different sites. Therefore, while ADAM17 enhances the shedding of *ACE-2* and increasing soluble *ACE-2*, TMPPRSS2 interferes with this and prevents the shedding of *ACE-2* (Figure 4). These findings suggest that ADAM17 is the major shedding enzyme for *ACE-2*, and ADAM17-shedded *ACE-2* (circulating *ACE-2*) protects lungs from the viral infection. While there is evidence for TMPRSS2 expression inhibiting the ADAM17-regulated shedding of *ACE-2*, it is not clear how TMPRSS2 competes with ADAM17 in cleaving *ACE-2* during SARS-CoV or SARS-CoV-2 infections. Even though SARS-CoV/SARS-CoV-2 can enter a target cell via two ways, which are a fusion of viral membrane and endocytosis (Figure 2), viral membrane fusion is 100 times more efficient in terms of viral replication. Therefore, even though ADAM17-regulated ectodomain shedding of *ACE-2* could induce viral entry through endocytosis, this would not be as effective as SARS-CoV/SARS-CoV-2 cell entry through fusion regulated by TMPRSS2, for which ADAM17 is not needed [12].

### 2.5. ACE-2 and ARDS

The lung itself can be very prone to viral infections, more than any other organ, since type II alveolar epithelial cells produce 83% of the cells that express *ACE-2* in the human body [30]. *ACE-2* is a crucial component of the renin–angiotensin–aldosterone system (RAAS) [17]. The RAAS system is significant in the homeostasis of respiratory and cardiovascular systems [31]. *ACE-2* converts ANG-II into the protective lung ANG-(1–7) and ANG-(1–9) through a pathway, catalyzing and preventing Angiotensin type I receptor (AT1R) activation, and, thus, *ACE-2* function in the opposing manner to counterbalance ACE [32]. It has been previously shown by Papp et al. [33] that AT1R is a probable cause for apoptosis in lung alveolar epithelial cells in response to ANG-II in alveolar epithelial cells in rats and humans. In addition, AT1R is highly expressed and it mediates signaling effects in pulmonary and cardiac tissues, causing inflammation, pro-fibrotic signaling, and tissue remodeling [34]. In addition, ANG-II has the capacity to release pro-inflammatory cytokines, resulting in eventually what is known as cytokine storm. It was also shown by Ismael-Badarneh that ANG-II decreases the rate of the clearance of alveolar fluids. Overall, it can be negative for pulmonary tissues, in particular, and a great factor in the injuries, hence the role of *ACE-2* arises as a protective enzyme for lungs [35].

In their pioneering work, Imai and colleagues showed that inducing ARDS in wild-type mice resulted in a dramatic reduction in lung *ACE-2* expression, suggesting *ACE-2* downregulation might be causing the severe acute lung pathologies in SARS-CoV [36]. Knocking out *ACE-2* almost completely attenuated acute lung injury. Importantly, injection of recombinant human *ACE-2* into acid-treated *ACE-2* knockout mice as well as to the wild-type mice decreased the degree of acute lung injury significantly. In their follow-up work, the same group investigated lung *ACE-2* expressions and ARDS in SARS-CoV infected mice [37]. Both *ACE-2* knockout and control wild-type mice were infected with SARS-CoV. Similar to previous reports, when wild-type mice were infected, large amounts of the infectious virus could be recovered from the lungs. However, when the *ACE-2* knockout mice were infected, only a very low quantity of virus could be extracted from the lung. Interestingly, a viral infection of the wild-type mice caused mild pathological changes in the lungs and decreased pulmonary *ACE-2* expression. For the knock-out mice, pathological alterations following viral infection were reduced compared to wild-type mice. To confirm whether SARS-CoV S-protein could affect the severity of acute lung injury, they treated both wild-type and knock-out animals with SARS-CoV S-protein. Similarly, spike protein administration worsened acid-induced acute lung injury in wild-type mice, but did not influence the severity of lung injury in knock-out animals, showing that the effect of S-protein on acute lung injury depends on *ACE-2*. These findings suggest, not the viral replication within the cell but the inactivation of *ACE-2* following the docking of the spike protein is the main factor for lung injury in SARS-CoV/SARS-CoV-2 infections. To further demonstrate the beneficial effect of *ACE-2* expression in ARDS, Li and colleagues generated a pulmonary *ACE-2* over-expression model by Lentiviral *ACE-2* cDNA or *ACE-2* shRNA administration into the lungs of SD rats [38]. The animals were exposed to LPS to induce ARDS. *ACE-2* overexpression significantly prevented LPS-induced lung injury and inflammatory response. In contrast to this finding, Pinto and colleagues examined more than 700 lung samples from severe COVID-19 patients who suffered from several comorbidities, and revealed high pulmonary *ACE-2* expression in these patients, compared to healthy individuals [39].

The mechanism of shedding cannot be thought of as an entirely understood mechanism, as it has been gaining a wider attention in order to become understood, especially with the emergence of COVID-19 and the increasing interest in the role of *ACE-2*. It has been suggested that the shedding of *ACE-2* might trigger pro-inflammatory response [40,41]. *ACE-2* shedding seems to be more inducible through two means: (1) viral infection response, and (2) immunological reaction [42]. The former was clearly understood with the emergence of SARS-COV in 2002–2003 [23]. As for the latter, it was suggested after the research of Jia et al., which manifested difference in primary airway epithelial cells and Calu-3-cell line due the *ACE-2* shedding [15].

As might be understood, ADAM17 plays a crucial role in *ACE-2* shedding into s*ACE-2*. The cleavage of membrane anchor occurs through ADAM17, and it underlies its occurrence in body fluid as the AT receptor upregulates ADAM17, resulting in an increase in s*ACE-2,* resulting in a decrease in SARS-COV-2 entry as the virus will interact with the soluble form instead of interacting with m*ACE-2* and invade the cells [43]. In addition, it is worth highlighting that the cleavage of m*ACE-2* occurs in a region called juxtamembrane, and its retention on the cell membrane is regulated by the binding of calmodulin, and as the calmodulin is inhibited, the sc*ACE-2* is increasingly released [44]. In addition, Imai et al. found that *ACE-2* has the capacity to cleave the C-terminal residue from Apelin [45]. This is crucial as Apelin’s receptor and agonist APJ and agonist apelin-13 protect the lung through the mechanism of reducing ANG-II.

In 2012, El-Hashim et al. showed that *ACE-2* may reduce the inflammations in the lung via what is known as ANG 1–7/MasR axis. In their research model of murine asthma, they found that it reduced eosinophil (Figure 5) [46]. They used a mouse model for allergic asthma and they assisted lungs to quantify goblet cell, perivascular, and peribronchal inflammation and fibrosis. In addition, they made a biochemical analysis of ERK1/2 and IκB-α. Furthermore, the effect of Ang-(1–7) on the proliferation of human peripheral blood mononuclear cells (HPBMC) was investigated. In their results, they found that Ang-(1–7) attenuated ovalbumin-induced increase in the otal cell counts, eosinophils, lymphocytes and neutrophil. On the other hand, decreased the ovalbumin-induced perivascular and peribronchial inflammation, fibrosis and goblet cell hyper/metaplasia, as well as decreasing the ovalbumin-induced increase in the phosphorylation of ERK1/2 and IκB-α. The authors concluded that ANG 1–7 via MAS1 receptor acts as an ant-inflammatory pathway in allergic asthma [47]. Reddy et al. demonstrated that in ARDS patients, it was found that the ratio of ANG 1–7 was higher compared to ANG-I among survivors [48].

Recently, clinical investigators examined seven lungs obtained during autopsy from patients who died from COVID-19 and compared them with seven lungs obtained during autopsy from patients who died from ARDS secondary to influenza A (H1N1) infection and 10 age-matched, uninfected control lungs. In their findings, they found a significant increase in the numbers of ACE2-positive cells in the lungs from patients with Covid-19 and from patients with influenza than in those from uninfected controls. In their results, they found that a great positive number of endothelial ACE2-positive, as well as changes in endothelial morphology [49].

The analysis of ACE2 expression in lungs reveals that it is very low and limited to a small fraction of type II alveolar epithelial cells (around 1%) [50] and, surprisingly, in both mRNA and protein levels [51], other types of cells release traces amounts [30,52]. As stated earlier, there are four types of isoforms of ACE2: (1) membrane bound or full length mACE2; (2) Soluble ACE2 sACE2; (3) Isoform 3 of ACE2, known as cACE2 or isoform 3; and (4) newly identified dACE2 [53]. Interestingly enough, it has been recently discovered that it is majorly located and expressed in differentiated airway epithelial cells, especially in cells of the upper airways. Previously, it was suggested that ACE2 is an interferon-simulated gene [54,55], but here the authors report that the discovery of the novel transcriptionally independent isoform-designated dACE2 is the interferon-simulated one, and no other isoforms. Perhaps this was not discovered earlier because 3’-scRNA-seq methods do not discriminate between ACE2 and dACE2. In their findings, the authors also found that normal primary bronchial respiratory cells are the baseline expression levels of dACE2, and they are further strongly induced by IFN treatments. This comes in agreement with Zhuang et al.’s speculations that AE2 can be upregulated by viruses or through the simulation of inflammatory cytokines, such as interferons [56].

In 2010, Wosten-van Asperen et al. investigated the role of ACE and ACE2 in animal models in LPS-induced ARDS. The animals were treated with losartan (ANG-II receptor antagonist), or with a protease-resistant, cyclic form of Ang-(1–7) [cAng-(1–7)]. In their results, they found that in treated rats that are LPS-exposed as well, BALF ACE2 activity and Ang-(1–7) levels were significantly higher in both treatment groups compared with the placebo group [57]. In 2020, Kimura et al. demonstrated whether IL-13 (a potent mediator of type 2 inflammation) modulates ACE2 and TMPRSS2 in airway epithelial cells. They determined the effects of IL-13 on ACE2 and TMPRSS2 expression in vivo in primary airway epithelial cells, from participants that have type 2 asthma. In their results, they found a significant role of IL-13 was to reduce ACE2 and increase TMPRSS2 expression in epithelial and nasal airways. In addition, they found out that ACE2 was negatively associated with type 2 cytokines. On the other hand, TMPRSS2 was positively associated with type 2 cytokines. *ACE-2* and TMPRSS2 are therefore modulated by type 2 inflammation in the upper and lower airways. This was found to be important for asthma and atopy setting in COVID-19 infection, since SARS-CoV-2 cell entry depends on *ACE-2* and TMPRSS2 [58].

Liu and colleagues showed that the Angiotensin-II level in the plasma sample from SARS-CoV-2-infected patients was significantly elevated and linearly associated with viral load and lung injury, suggesting decreased *ACE-2* expression and function in these patients, consistent with previous animal work on SARS-CoV [59]. Similar to studies for SARS-CoV, a previous study on influenza A (H7N9) demonstrated that circulating *ACE-2* protects the lung against virus-induced acute lung injury [60]. Knocking out *ACE-2* in mice led to severe lung damage when the animal was infected with H5N1, and treatment of these animals with human *ACE-2* attenuated the lung damage. These findings provide strong evidence for the protective roles of soluble as well as pulmonary *ACE-2* against non-viral ARDS and viral ARDS from SARS-CoV-2 [61].

Sodhi et al. investigated the effects on des-Arg^9^ bradykinin (DABK) in airway epithelial cells based on a hypothesis that DABK is a biological substrate of ACE2 in the lung, and ACE2 has a significant role in the pathogenesis of acute lung inflammation through modulating DABK/bradykinin receptor B1 (BKB1R) axis signaling. In their results, they found that ACE2 function in the lungs of the mice in the setting of endotoxin inhalation led to the activation of the DABK/BKB1R axis releasing pro-inflammatory chemokines, such as C-X-C motif chemokine 5 (CXCL5), macrophage inflammatory protein-2 (MIP2), C-X-C motif chemokine 1 (KC), and TNF-α from airway epithelia, increased neutrophil infiltration, and exaggerated lung inflammation and injury. Their results gave a greater insight that the reduction in pulmonary ACE2 activity increase the pathogenesis of the lungs inflammation, as the authors indicated that this in part because of an impaired ability to inhibit DABK/BKB1R axis-mediated signaling and this results in more neutrophil infiltration and more severe inflammation in the lung [62]. Authors successfully demonstrated that rhACE2 improves pulmonary blood flow and oxygenation in lipopolysaccharide-induced lung injury in the piglet’s model. In their results, they found that animals treated with rhACE2 were maintaining higher partial oxygen pressure (PaO_2_) compared to animals that were in the control group. In addition, they noticed that pulmonary hypertension was less pronounced. They also noticed that inflammatory markers, such as ANG-II and tumor neurosis factor-alpha, that were subsequently increased, were returned to basal values [63]. The rhACE2 they used was isolated and extracted in the laboratory. In 2017, Khan et al. conducted a pilot clinical trial (NCT01597635) to test the safety of rhACE2 in ARDS patients. In their trial, they used GSK2586881, which is an rhACE2, to investigate attenuating acute lung injury. In their trial, they concluded that infusion of patients with the rhACE2 caused changes in RAS biomarkers. In addition, they found that it is tolerated well in patients with ARDS [64].

### 2.6. ACE-2 and CVDs

*ACE-2*’s main function is to counterbalance ACE and ANG-II as part of RAS. Cardiac *ACE-2* expression, as well as soluble *ACE-2* levels, are altered during CVDs, and these altered expressions are closely related to cardiac function. Since CVDs are common among COVID-19 patients, it becomes important to reveal the possible effects of altered *ACE-2* mechanism in SARS-CoV-2 virulence and cardiac injury for these patients. Most of the relevant previous studies focused on the soluble and cardiac *ACE-2* levels in diseased states, aiming to understand how each of these expressions influences cardiac health. In one of the early works, Donoghue and colleagues generated transgenic mice with increased cardiac-specific *ACE-2* expression [65]. The animals had a high incidence of sudden death that correlated with *ACE-2* expression levels. Serum *ACE-2* enzymatic activity was higher in transgenic animals compared to non-transgenic littermates, indicating that soluble *ACE-2* was released into circulation. Interestingly, *ACE-2* transgene was downregulated in surviving older animals and cardiac function was restored. The study is important in showing that increased tissue *ACE-2* expression correlates with increased soluble *ACE-2*. However, other than that, transgenically increasing cardiac *ACE-2* in otherwise healthy animals is not clinically relevant. In a more clinically relevant study, Huentelman and colleagues showed that lentiviral induced cardiac overexpression of *ACE-2* protects the heart from ANG-II-induced hypertrophy and fibrosis in rats [66,67,68,69,70,71,72]. Crackower and colleagues showed that targeted disruption of *ACE-2* in mice results in a severe abnormal cardiac contractility, increased ANG-II levels, and upregulated hypoxia-induced cardiac gene expression [73]. Consistent with that finding, cardiac myocardial *ACE-2* expression significantly increases in failing cardiomyopathic hearts compared to healthy controls [74], suggesting the need for an active compensatory mechanism in RAS to preserve heart function for these patients. More recently, Bos and colleagues also demonstrated a 5-fold increase in myocardial *ACE-2* expression for obstructive hypertrophic cardiomyopathy patients [4]. Similar findings were presented by Chen and colleagues at both gene and protein levels for basic heart failure [75].

Regarding soluble *ACE-2*, interestingly, Epelman, and colleagues showed that increased *ACE-2* plasma activity strongly correlates with a heart failure diagnosis, compared to healthy individuals [76]. In a following study, the same group measured soluble *ACE-2* activity in patients with chronic systolic heart failure, and showed that elevated plasma *ACE-2* activity is associated with more severe myocardial dysfunction [77]. Increased soluble *ACE-2* levels were observed in a variety of other CVDs as well. Soro-Paavonen and colleagues revealed increased circulating *ACE-2* activity in patients suffering from micro- and macrovascular complications associated with type 1 diabetes [78]. Walters and colleagues examined plasma *ACE-2* enzyme levels for atrial fibrillation patients, and showed that increased plasma *ACE-2* activity is associated with atrial fibrillation and more advanced left atrial structural remodeling [79]. Li and colleagues revealed elevated serum *ACE-2* concentration in hypertension patients compared to in healthy individuals [80]. Soluble *ACE-2* levels correlated well with cardiac remodeling as found from left atrial diameter, left ventricular end-diastolic diameter, and left ventricular mass for hypertensive patients. These findings demonstrated increased soluble *ACE-2* in a variety of CVDs, suggesting that a cardioprotective arm of the RAAS system is active in these conditions. Since none of these studies examined soluble and cardiac *ACE-2* levels simultaneously, it is not clear whether elevated soluble *ACE-2* activity is due to increased expression of tissue *ACE-2* and then shedding into the circulation, or the elevated circulating *ACE-2* is a result of a loss of protective *ACE-2* from the tissue, exacerbating the cardiac injury.

In the context of COVID-19, it is not clear how SARS-CoV-2 infection affects *ACE-2* levels for already abnormal expressions due to CVDs, and whether this plays a role in virulence and cardiac complications. The effect of SARS-CoV-2 infection on the circulatory system other than the lung is not established [81]. The myocardial dysfunction in COVID-19 can be indirect, as a consequence of severe lung injury and reduced oxygen supply, or cytokine storm following the SARS-CoV-2 infections [82]. The effect can also be direct, due to decreased *ACE-2* expression and activity in the heart following myocardial infection. Oudit and colleagues studied whether SARS-CoV can infect myocardial tissue to cause myocarditis [83]. For wild-type mice, following the pulmonary infection, SARS-CoV was present in the heart. Interestingly, virus levels in the heart were reduced in *ACE-2* knockout mice, suggesting myocardial infection is *ACE-2*-dependent. Myocardial infection in wild-type mice also induced partial *ACE-2* mRNA downregulation and a complete loss of myocardial *ACE-2* protein. In the same study, the authors analyzed the SARS-CoV virus and *ACE-2* expression in autopsied heart samples from patients who died from the infection. SARS-CoV virus and myocardial inflammation were present in seven out of 20 samples analyzed. Similar to animal experiments, SARS-CoV was present in the myocardial tissue, and myocardial *ACE-2* protein expression was markedly downregulated. Similar investigations are needed for SARS-CoV-2 to reveal whether the virus is capable of causing myocardial infection. Myocarditis is not common among COVID-19 patients, and even for a few reported cases, it is not clear whether the infection is caused by the SARS-CoV-2 virus [84]. Endothelial dysfunction and inflammation, on the other hand, seem to be common among COVID-19 patients, as clearly seen from the high incidence of thrombotic complications [85]. In addition to the thrombotic events, central and peripheral nervous manifestation are frequent in patients with COVID-19 [86]. However, even for these cases, it is not clear whether these complications are caused by a direct viral infection or are secondary to other complications.

It is most likely that cardiac complications due to SARS-CoV-2 are due to altered *ACE-2* mechanism. Several current studies suggest that increasing particularly soluble *ACE-2* is a logical approach for protection against SARS-CoV-2-related cardiac complications. Administrating recombinant human *ACE-2* to pulmonary arterial hypertension patients demonstrated hemodynamic improvement [87]. One interesting fact is that women have increased serum *ACE-2* compared to men, and women seem to have milder symptoms of COVID-19. In addition, the aging population seems to have less soluble *ACE-2*. Again, children usually present milder symptoms and are asymptomatic for SARS-CoV-2 infection [61]. A correlation seems to exist between age and severity of COVID-19. This correlation may not only be due to the declining immune system for the elderly, but also be because of a specific plasma *ACE-2* profile for children. Therefore, in addition to hemodynamic benefits, circulating *ACE-2* may be helping children, women, and asymptomatic COVID-19 patients to better deal with the virus by buffering it in the circulation similar to neutralizing antibodies. Human recombinant soluble *ACE-2* is suggested as a feasible approach to decreasing the severity of viral infection and cardiovascular improvement. In that context, Monteil and colleagues showed that clinical-grade human recombinant soluble *ACE-2* can efficiently prevent SARS-CoV-2 viral infection and growth for Vero E6 cells [88]. They also showed that viral growth within SARS-CoV-2-infected human blood vessel organoids and kidney organoids can be significantly inhibited via administration of soluble *ACE-2* at the early stage of infection.

### 2.7. Altered ACE-2 Expression with Certain Medications: Is There Evidence of Altered Expression and SARS-CoV-2 Virulence?

As summarized in the previous section, *ACE-2* overexpression can prevent and even reverse heart failure phenotype. *ACE-2* downregulation, on the other hand, can accelerate the progression of heart failure [82]. Relevant to COVID-19, *ACE-2* expression, and susceptibility to SARS-CoV infection seem to correlate well from in vitro studies [22]. However, up to date, no direct in vivo evidence has been provided for such a correlation. Nevertheless, following the discovery of the SARS-CoV-2 virus utilizing *ACE-2* for cell entry, significant attention has been directed to *ACE-2*, and to clinical drugs that are thought to affect *ACE-2* levels.

Hypertension is among the most common comorbidities for COVID-19 patients. ARBs and ACEi are the most commonly prescribed class of medication for hypertension. In addition to their main pharmacological effect to block the ANG-II type 1 receptor or to inhibit ACE, interestingly, several studies have suggested that these drugs may upregulate the expression of *ACE-2* as well [82] (Figure 6). Accepting that *ACE-2* expression correlates with the susceptibility to SARS-CoV-2 infection, one would assume that ACEis/ARBs might increase the risk of SARS-CoV-2 infection. But is this the case? To answer this question, first, it should be known whether ACEis/ARBs increase *ACE-2* levels in clinical settings. Then, evidence should be provided if increased *ACE-2* increases virulence.

Akhtar and colleagues have recently reviewed the studies on clinical drugs affecting *ACE-2* levels [89]. The authors confirmed that most relevant studies were on animals, and these showed that both ACEi and ARBs increased circulating as well as cardiac *ACE-2* levels. Assuming these medications increase cardiac *ACE-2* in clinical settings in a similar manner to animal experiments, and also increased cardiac *ACE-2,* would mean more viral infection in the heart, and SARS-CoV-2-related myocarditis would be a common complication for COVID-19 patients under ACEi/ARB. However, other than a few reports, this is not the case [84]. So far, there are only a few epidemiological studies for ACEi/ARB medication for COVID-19 patients. Xiao and colleagues assessed epidemiology and disease severity for hypertensive COVID-19 patients under ACE/ARB, and compared with patients under other hypertension medication [90]. Even though the cohort size was small, with 42 patients, the authors confirmed that patients under ACEi/ARB therapy had a lower incidence of severe diseases and also lower levels of IL-6 in peripheral blood, supporting the benefit of using ACEis or ARBs for improving the clinical outcomes of hypertensive COVID-19 patients.

While originally it was speculated that ACEi/ARBs-related upregulation of *ACE-2* would result in more severe disease outcomes for COVID-19 patients, conversely, most current studies show that ACEi/ARB either has a neutral or beneficial effect for COVID-19 hypertensive patients in comparison. Guo and colleagues performed a meta-analysis of the current studies to explore whether the use of ACEi/ARB was associated with disease severity and mortality in COVID-19 patients with hypertension [91]. An analysis of nine studies involving 3936 patients showed that ACEi/ARB therapy did not aggravate disease severity, and it could even decrease the mortality of COVID-19 patients. For a cohort of 205 patients, Bean and colleagues showed that ACEi treatment was associated with reduced severe disease risk [92]. Salah and colleagues analyzed 12 studies involving 16,101 patients, and showed that the use of ACEi and ARBs improves mortality for hypertensive COVID-19 patients, with similar outcomes between ACEis and ARBs in the studied population [93]. Gao and colleagues examined 2877 patients admitted to Huo Shen Shan Hospital, a hospital dedicated to the treatment of COVID-19 in Wuhan, China [94]. The analysis revealed that the use of ACEi/ARB is associated with a lower risk of mortality. Mehra and colleagues performed a similar observational study for 169 hospitals in Asia, Europe, and North America, and studied the relationship of CVDs and drug therapy with in-hospital death among hospitalized patients with COVID-19 [95]. Authors found no increased risk of in-hospital death with the use of ACEi or ARBs. Zhang and colleagues performed a multi-center retrospective study including 1128 adult patients with hypertensive COVID-19 patients, and showed that inpatient use of ACEI/ARB was associated with a lower risk of all-cause mortality compared to ACEi/ARB non-users [96].

To sum up, present evidence suggests that ACEi/ARBs may increase the expression and activity of cardiac *ACE-2* as well as soluble *ACE-2*, performing a protective role in the cardiovascular system, as seen clearly from several meta-analyses. However, the effect of these drugs on *ACE-2* in other organs, in particular, whether they could affect the expression and activity of pulmonary *ACE-2*, remains unknown. Assuming ACEi/ARBs may upregulate the expression and activity of *ACE-2* in the lungs, this may have a protective influence against acute lung injury due to SARS-CoV-2 infection, as explained in the previous section [82]. Alternatively, Exogenous supplement of recombinant human *ACE-2* is a useful approach in the treatment of COVID-19, especially for those patients with cardiovascular diseases. Increasing *ACE-2* levels with *ACE-2*-increasing medication is another approach that needs further exploration (Figure 7). Furthermore, recently, Inchingolo and colleagues revealed that improving immune system along with sartans as adjuvant therapy potentially prevented COVID-19 infection [97].

## 3. Conclusions

As the host receptor for SARS-CoV-2, *ACE-2* is a very important enzyme for COVID-19. Viral entry into *ACE-2*-expressing cells is via viral S-protein, which is mainly conserved between SARS-CoV and SARS-CoV-2. While previous investigations are mostly on SARS-CoV, infection mechanisms can be assumed to apply to SARS-CoV-2 due to the high resemblance of S-proteins RBD domains. Even though *ACE-2* is expressed in a variety of different tissues throughout the body, the lung is the primary target for SARS-CoV-2. There is no evidence for increased susceptibility to viral infection due to an *ACE-2* upregulation for lung as well as for other *ACE-2*-expressing tissues. Conversely, viral infection decreases pulmonary *ACE-2*, which is the primary cause of ARDS and lung complications in COVID-19. In addition to its role in viral infection, *ACE-2* has an important function in RAAS to control ANG-II levels. Disruption of the *ACE-2* mechanism seems to be the primary cause of adverse cardiac events in COVID-19. Increasing both soluble and tissue-specific (lung and cardiac) *ACE-2* levels will have a protective influence against COVID-19 and related complications.

## Figures and Tables

**Figure 1 ijms-22-06703-f001:**
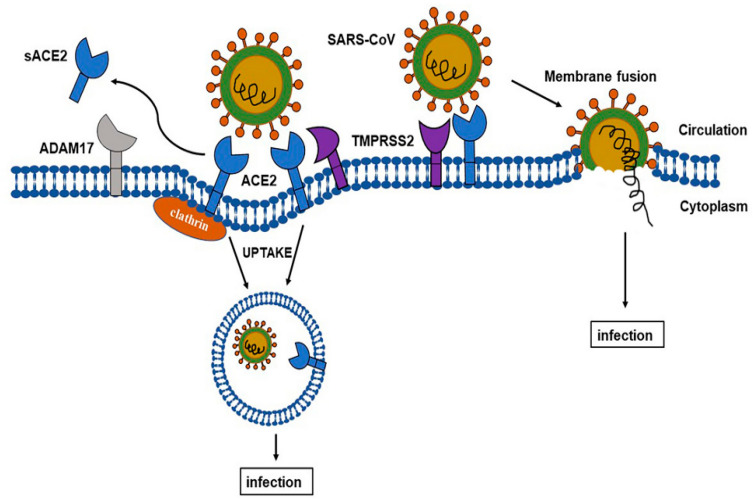
SARS-CoV’s viral invasion of the host cell via a membrane-bound *ACE-2* receptor. *ACE-2* mediates entry of coronavirus in two distinct ways. The first one is Clathrin and ADAM 17-dependent endocytosis, and the second one is TMPRSS2-dependent membrane fusion. Figure replicated from [6] with permission.

**Figure 2 ijms-22-06703-f002:**
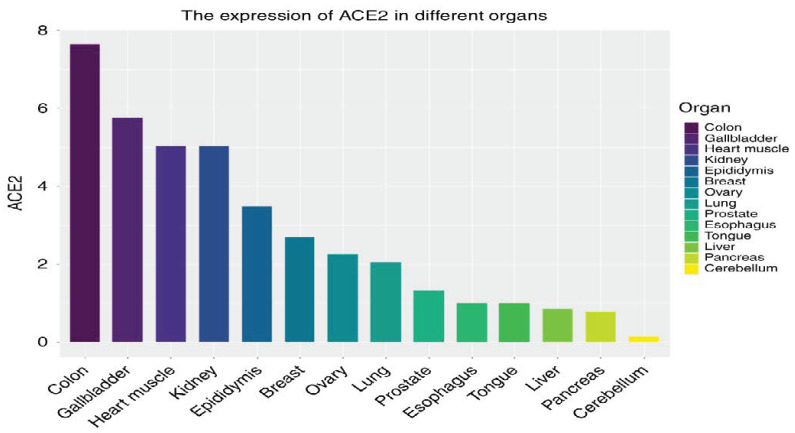
Organ-specific comparative *ACE-2* expression. Figure replicated from [13] with permission.

**Figure 3 ijms-22-06703-f003:**
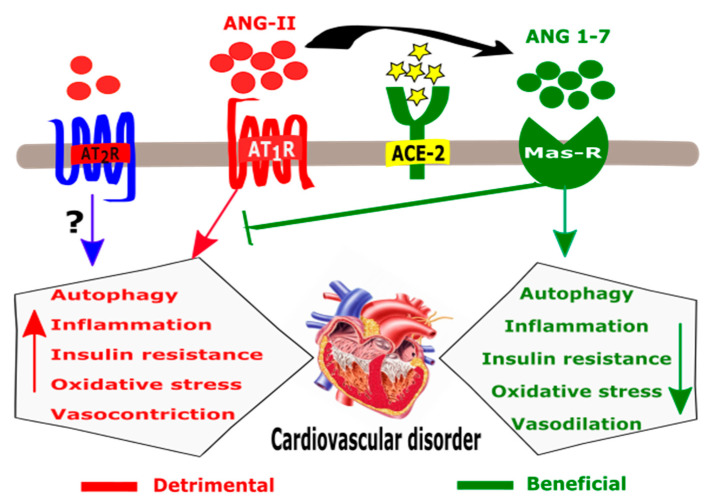
Schematic diagram for the major pathways in RAAS.

**Figure 4 ijms-22-06703-f004:**
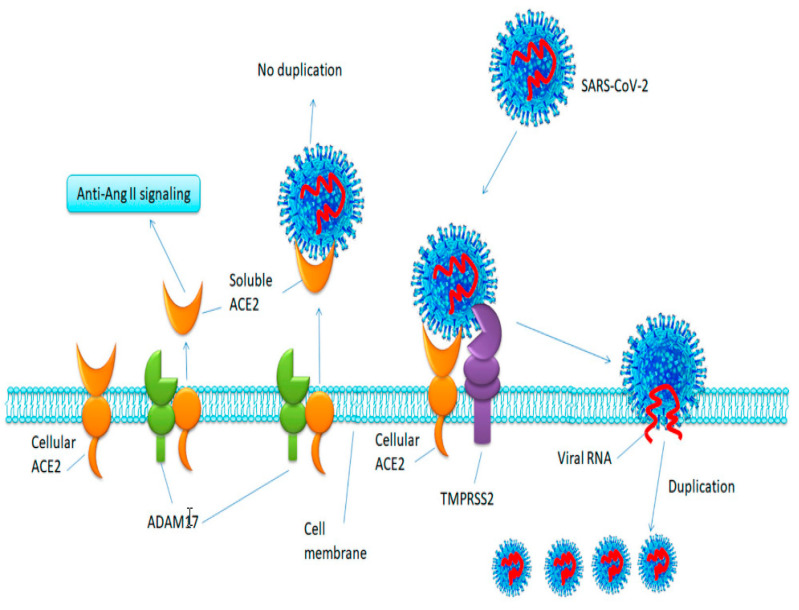
*ACE-2* shedding and SARS-CoV-2 entry. TMPRSS2 process *ACE-2* for efficient viral entry and replication. Figure replicated from [12] with permission.

**Figure 5 ijms-22-06703-f005:**
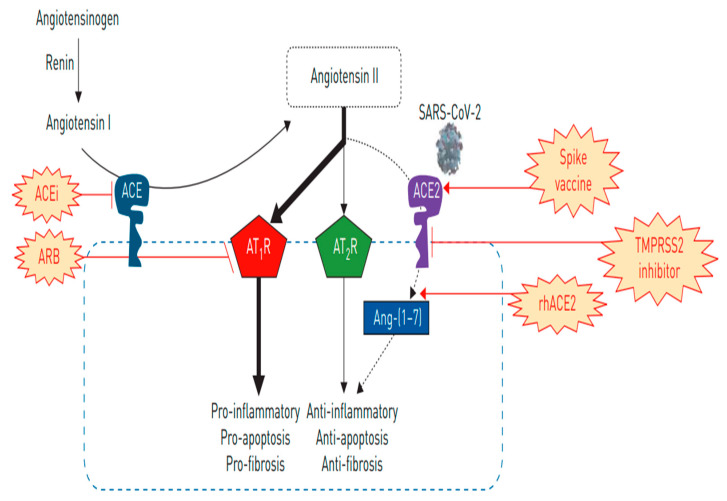
The RAAS with COVID-19. The thicker arrows show an increase in the degree of pathway activation; dotted arrows show a decrease in pathway activation. ACE: angiotensin-converting enzyme; *ACE-2*: angiotensin-converting enzyme-2; AT1R: angiotensin type 1 receptor; AT2R: angiotensin type2 receptor; ANG 1–7: angiotensin 1–7; ARB: angiotensin receptor blockers; rh*ACE-2*: recombinant human *ACE-2*; TMPRSS2: transmembrane serine protease 2 (figure replicated from [46] with permission).

**Figure 6 ijms-22-06703-f006:**
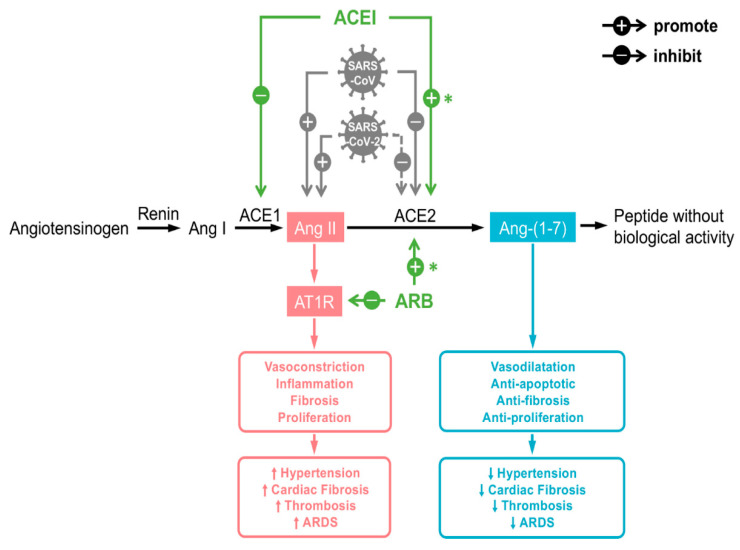
Working mechanisms for ACEi and ARB and relevance to SARS-CoV/SARS-CoV-2 infections. Figure replicated from [82] with permission.

**Figure 7 ijms-22-06703-f007:**
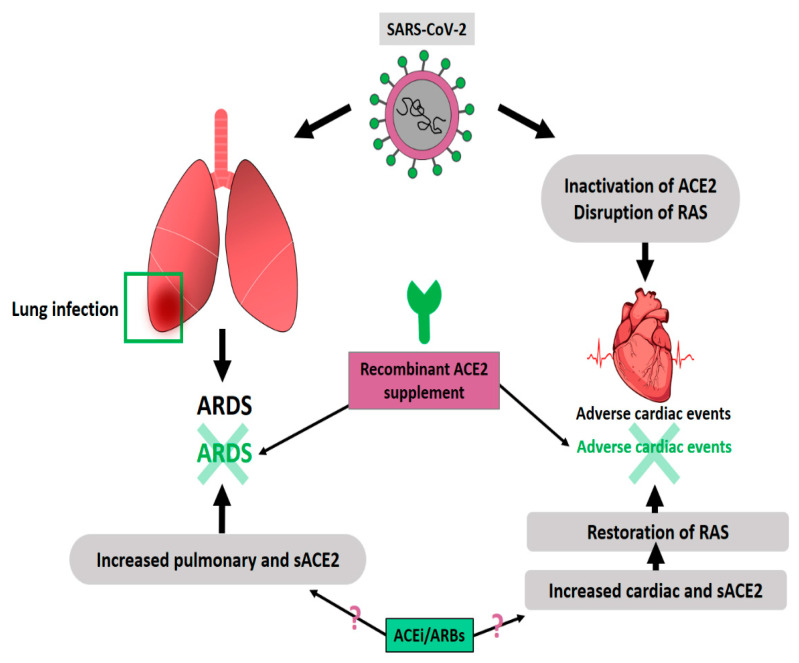
Potential approaches against ARDS and adverse cardiac events relevant to SARS-CoV-2 infection.

## Data Availability

All experimental data to support the findings of this study are available contacting the corresponding author upon request.

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
