# Peer review of "Do Changes in ACE-2 Expression Affect SARS-CoV-2 Virulence and Related Complications: A Closer Look into Membrane-Bound and Soluble Forms"

_ijms, 2021, doi:10.3390/ijms22136703_

Round 1

Reviewer 1 Report

An important and timely review. Overall quite well-structured and informative. 

Minor comment

  1. The review contains figures adapted from other articles - which were clearly indicated as such. Please make sure approval from the authors of the original articles & their publishers have been obtained - otherwise, there could be serious repercussions if they filed complaints on copyright/plagiarism grounds!
  2. This issue also shows up as language issues "imported" from those articles. For example, in Figure 4, duplication should read replication
  3. I suggest following the official HGNC Approved Gene Symbol  (and apply Italics as necessary) for gene names such as ACE2. Also see https://www.omim.org/entry/300335?search=ACE2&highlight=ace2
  4. Some citations are incomplete, e.g. journal name is missing in 12. Xiao L, Sakagami H. ACE2: The key Molecule for Understanding the Pathophysiology of Severe and Critical Conditions of COVID-19: Demon or Angel? 2020;12(5).  The reference list needs to be carefully checked. 
  5. Minor stylistics issues and typos. 
    1. It’s thought that the shedding of ACE-2 itself can trigger proinflammatory response (40) 
    2. 2020, Ackermann examined 7 lungs obtained during autopsy from patients who died from Covid19

Author Response

Reviewer 1

Response: We sincerely thank reviewer 1 for their comments and recommendations. We hope that this revision will be acceptable.  

  1. The review contains figures adapted from other articles - which were clearly indicated as such. Please make sure approval from the authors of the original articles & their publishers have been obtained - otherwise, there could be serious repercussions if they filed complaints on copyright/plagiarism grounds!

Response: We have adapted the figures with permission. We got the approval from the authors and journal publishers.

  1. This issue also shows up as language issues "imported" from those articles. For example, in Figure 4, duplication should read replication

Response: We have revised (page 03, line 120; page 05, line 153; page 8, line 262; page 10, line 336 and page 15 line 534).

  1. I suggest following the official HGNC Approved Gene Symbol  (and apply Italics as necessary) for gene names such as ACE2. Also see https://www.omim.org/entry/300335?search=ACE2&highlight=ace2

Response: We have revised as ACE-2 (italics) throughout the manuscript.

  1. Some citations are incomplete, e.g. journal name is missing in 12. Xiao L, Sakagami H. ACE2: The key Molecule for Understanding the Pathophysiology of Severe and Critical Conditions of COVID-19: Demon or Angel? 2020;12(5).  The reference list needs to be carefully checked. 

Response: We have corrected this reference (page 19, line 662) and verified the remaining references.

  1. Minor stylistics issues and typos. 
  2. It’sthought that the shedding of ACE-2 itself can trigger proinflammatory response (40) 
  3. 2020, Ackermann examined 7 lungs obtained during autopsy from patients who died from Covid19

Response: We have revised the texts (page 9 line 313 and page 11, line 351).

Reviewer 2 Report

The article has to be inserted on the template of the journal because it is not inserted. It is in a word document.

Please rephrase these phrase "2020, Ackermann examined 7 lungs obtained during autopsy from patients who died from Covid-19 and compared them with 7 lungs obtained during autopsy from patients who died from ARDS secondary to influenza A (H1N1) infection and 10 age-matched, uninfected control lungs." because you can not start a phrase with a year.

You can not insert a figure in the conclusion "Exogenous supplement of recombinant human ACE-2 is a useful approach in the treatment of COVID- 19, especially for those patients with cardiovascular diseases. Increasing ACE-2 levels with ACE-2 increasing medication is another approach that needs further exploration (Figure 7). " Please rephrase.

The bibliography has to be formated as indicated by the journal.

Please update the bibliography. You could also cite this articles.

DOI:10.3390/microorganisms8111704

DOI:10.3390/ijerph17218049

DOI:10.2147/RMHP.S284557

DOI:10.3390/microorganisms9040793

DOI:10.3390/microorganisms9030525

Author Response

Reviewer 2

Response: We are grateful for the insights of the reviewer. We have revised the manuscript extensively. We have tried our best in revision to accommodate your suggestions and hope this revision will be acceptable.

The article has to be inserted on the template of the journal because it is not inserted. It is in a word document.

Response: The revised manuscript prepared as per the journal format (IJMS template used).

Please rephrase these phrase "2020, Ackermann examined 7 lungs obtained during autopsy from patients who died from Covid-19 and compared them with 7 lungs obtained during autopsy from patients who died from ARDS secondary to influenza A (H1N1) infection and 10 age-matched, uninfected control lungs." because you cannot start a phrase with a year.

Response: We have revised the texts (page 9 line 313 and page 11, line 351).

You can not insert a figure in the conclusion "Exogenous supplement of recombinant human ACE-2 is a useful approach in the treatment of COVID- 19, especially for those patients with cardiovascular diseases. Increasing ACE-2 levels with ACE-2 increasing medication is another approach that needs further exploration (Figure 7). " Please rephrase.

Response: We have rephrased the sentences and figure 7.

The bibliography has to be formatted as indicated by the journal.

Please update the bibliography. You could also cite this articles.

DOI:10.3390/microorganisms8111704

DOI:10.3390/ijerph17218049

DOI:10.2147/RMHP.S284557

DOI:10.3390/microorganisms9040793

DOI:10.3390/microorganisms9030525

Response: We have formatted the references as per the IJMS journal and adapted recent articles.

Round 2

Reviewer 2 Report

Now in my oppinion the article is suitable for publication.